# OpenReview forum: "ShoppingComp: Are LLMs Really Ready for Your Shopping Cart?"
_ICLR.cc/2026/Conference — ICLR 2026 Conference Desk Rejected Submission_

### Official Review · Reviewer_JBSD · 2025-11-01

**Soundness:** 3
**Presentation:** 3
**Contribution:** 3
**Rating:** 2
**Confidence:** 4

**Summary:**

Overall, I think the paper is good for writing and I have been enjoyed reading it. My judgement is that the paper might not meet the high standard of the papers at ICLR level.

**Strengths:**

A very good written paper

**Weaknesses:**

Actually the topic of this paper is in my research area. I have kind of feeling that the paper did not meet the quality as a ICLR paper.

**Questions:**

Would suggest authors do more in-depth analysis. Current one is too short and too shallow.

---

> ### Author Response · Authors · 2025-11-26
> **To Reviewer JBSD**
>
> >Q1: Would suggest authors do more in-depth analysis. Current one is too short and too shallow.
>
> We sincerely thank the reviewer for recognizing the paper’s writing quality. In response, we have added a new section: Section 4.4 “Resource and Tool-Usage Ablations” and revised Section 3.5 “Human Performance and Dataset Diversity”, which provide deeper and more systematic analyses of model behavior, dataset composition, and human evaluation.
>
> **1. Resource and Tool-Usage Ablations**
>
> This section (4.4 , pp. 10), examines how retrieval tools and token allocation influence real-world shopping performance.
> Table 4 reports F1, precision, recall, and resource usage (#calls and tokens) with and without tools. Key findings are as follows:
>
> - All models show extremely low recall (1–3%) without tools, confirming that parametric knowledge alone is insufficient for long-tail product retrieval.
>
> - Once tools are enabled, performance gains stem almost entirely from recall improvement, as models can retrieve and verify real-world evidence beyond pre-training data.
>
> - However, for most models, precision slightly decreases as broader retrieval introduces contextual noise; however, GPT-5 maintains or even improves precision by issuing structured searches and verifying key constraints against trusted sources.
>
> We further analyze retrieval strategies and efficiency differences: GPT-5 issues broader structured queries and allocates tokens to high-value evidence gathering, while Gemini-2.5-Pro relies on precision focused single calls. We have also added detailed tool-call trajectories in Appendix B.2 for this two models.
>
> | **Model**            | **F1 (w/o tool)** | **F1 (w/ tool)** | **P (w/o tool)** | **R (w/o tool)** | **P (w/ tool)** | **R (w/ tool)** | **#Calls (avg.)** | **Tokens (w/o tool)** | **Tokens (w/ tool)** |
> | :------------------- | :---------------: | :---------: | :------: | :--------: | :--------: | :-----: | :-------: | :-------: | :---------: |
> | **GPT-4o**           |        2.41       |       2.59       |       21.43      |       1.28       |       4.53      |       1.95      |        4.39       |         0.89k         |         6.53k        |
> | **GPT-5**            |      **4.63**     |     **11.22**    |     17.67    |     **2.66**     |    **19.40**    |     **7.97**    |     **20.42**     |       **5.42k**       |       **47.1k**      |
> | **Gemini-2.5 Flash** |        3.01       |       3.92       |       **34.52**     |       1.58       |      20.11      |       2.18      |        0.71       |         4.13k         |         5.99k        |
> | **Gemini-2.5 Pro**   |        2.21       |       7.73       |       20.99      |       1.17       |      15.92      |       5.12      |        2.50       |         2.02k         |         7.82k        |
> | **DeepSeek-V3.1**    |        3.09       |       5.71       |       28.26      |       1.64       |      18.63      |       3.39      |        7.37       |         4.99k         |         18.0k        |
>
> These analyses provide new insight into how retrieval strategy and evidence quality, rather than token count alone, determine real-world performance, highlighting ShoppingComp as a natural benchmark for studying retrieval augmented reasoning and efficient tool usage in complex, open-world settings.
>
> **2.Human Performance and Dataset Diversity**
>
> - *Dataset Diversity.* In Section 3.5, We also extended our analysis of dataset composition and human evaluation to provide greater interpretability.
> ShoppingComp consists of 120 tasks and 1,026 real-world scenarios, including 54 synthesized user tasks, 40 expert-authored tasks, and 26 safety-critical tasks. It follows Amazon’s hierarchical taxonomy, covering ten product categories such as Appliances, Electronics, Beauty & Personal Care, Health & Household, and Sports & Outdoors.
>
> - *Human Performance.*  To quantify task difficulty, we analyzed expert and annotator completion times.
> Experts spend an average of 24.6 minutes per case, while annotators spend 52.3 minutes, after capping outliers longer than one hour (experts) or two hours (annotators).\
> These results confirm the intrinsic difficulty of the benchmark and define a rigorous human upper bound for model evaluation.
> Even domain experts must perform multiple web searches to verify fine-grained product details, showing that ShoppingComp effectively tests retrieval, reasoning, and verification under realistic conditions.
>
> **Summary**
>
> Together, these new analyses provide much deeper insight into how tool use impacts score,  how models allocate resources, and how humans perform.
> We believe these substantial improvements significantly enhance the paper’s analytical depth and rigor, directly addressing the reviewer’s concern.
> We sincerely hope these revisions demonstrate our genuine effort to strengthen the work and may lead you to reconsider your evaluation.

---

### Official Review · Reviewer_dMUb · 2025-11-01

**Soundness:** 4
**Presentation:** 3
**Contribution:** 3
**Rating:** 6
**Confidence:** 3

**Summary:**

The authors propose a benchmark for evaluating the performance of LLM-driven shopping agents in real scenarios. It specifically evaluates the LLM's product retrieval ability, expert-level report generation ability, and the reliability of making safe decisions. In terms of evaluation dimensions, it expands on existing work's precise product retrieval and report generation quality by adding a new evaluation dimension for product safety assessment. Experiments on numerous tasks and scenarios reveal the performance limitations of various LLMs in real application scenarios and their discrepancies with human results. This research is interesting and valuable, providing a realistic and verifiable benchmark for evaluating e-commerce LLM agents.

**Strengths:**

1. This research is interesting and valuable. The authors propose a benchmark for evaluating LLM-driven shopping agents in real tasks and scenarios, which is no longer limited to academic benchmarks but focuses more on users' actual needs and shopping experience. This provides a real application direction and reference for related research fields.
2. Besides evaluating the LLM agent's product retrieval ability and report generation quality, the authors have also expanded with a safety decision reliability evaluation. It evaluates whether the products recommended by the LLM agent have safety hazards and false information, and assesses whether the LLM can accurately filter these non-safe products. This is also a key concern for users. The introduction of this evaluation metric also enhances the benchmark's evaluation completeness.
3. A new report evaluation metric has been proposed, enhancing the reliability of assessments in real-world scenarios.
4. Experiments reveal the performance differences of LLM-driven shopping agents between academic benchmarks and real scenarios, as well as the persistent lag behind human performance. This also provides valuable insights and considerations for the practical deployment of LLM agents.
5. The article's presentation is good and smooth. The introduction of the problem, analysis of related works, data collection, and experimental design are relatively clear.

**Weaknesses:**

1. There is a lack of introduction about the proposed benchmark containing 120 tasks and 1026 real scenarios, how this benchmark is composed, and how it is organized in subsequent testing. This point is not clearly explained.
2. The description of the data collection process in Section 3.2 is insufficient. Readers cannot understand the specific format and presentation of the data. Examples should be provided, for instance in the form of figures or tables. These can be provided in supplementary materials.
3. The same problem appears in Section 3.3. This section introduces two LLM verifiers used to check whether products meet specific scenarios. How exactly the checking is done, and what the fulfillment criteria are, lack detailed introduction. It is recommended to provide a detailed introduction to facilitate reader understanding.

**Questions:**

Please refer to weaknesses.

---

> ### Author Response · Authors · 2025-11-26
> **To Reviewer dMUb**
>
> We sincerely thank the reviewer for the encouraging and detailed comments. Below we address each concern point by point.
> >W1: There is a lack of introduction about the proposed benchmark containing 120 tasks and 1026 real scenarios, how this benchmark is composed, and how it is organized in subsequent testing. This point is not clearly explained.
> - We have added more detailed descriptions of the dataset composition in the revision. Specifically, the benchmark consists of 120 tasks covering 1,026 real-world scenarios, including 54 synthesized questions, 40 expert-generated questions, and 26 safety critical questions. Each task contains four core components: a question, a set of rubrics for evaluation, ground-truth products, and corresponding evidence to support judgments.
> - The dataset is being released on Hugging Face, with the code open-sourced on GitHub and a leaderboard planned on OpenReview.
>
> >W2: The description of the data collection process in Section 3.2 is insufficient. Readers cannot understand the specific format and presentation of the data. Examples should be provided, for instance in the form of figures or tables. These can be provided in supplementary materials.
>
> We sincerely appreciate the reviewer’s suggestion. To make the data collection process clearer, we have redesigned Figure 3 to illustrate the four-stage pipeline:
> 1. Question Collection : Includes synthesized, expert-authored, and safety-critical questions.
> 2. Rubric Generation : Experts and agents jointly create fine-grained rubrics.
> 3. Candidate Product Construction : Integrates retrieval results from web agents, expert-curated lists, and semantic search.
> 4. Ground-truth Verification and Filtering : Multiple stages of human annotation and expert cross-review ensure reliability and fairness.
>
> To further clarify the dataset format and presentation, two representative examples (“a microwave oven that cannot heat metal containers” and “selecting a monitor for a game live streamer”) are added in Appendix B.
> >W3. The same problem appears in Section 3.3. This section introduces two LLM verifiers used to check whether products meet specific scenarios. How exactly the checking is done, and what the fulfillment criteria are, lack detailed introduction. It is recommended to provide a detailed introduction to facilitate reader understanding.
>
> The revised version now includes full prompt templates and detailed explanations of the LLM verifiers in Appendix C. We summarize the LLM verifiers used in ShoppingComp in the following table:
> | **Prompts**                             | **Metric**                     | **Purpose**                                                                                                                         | **Input Format**                                           | **Output**                                                                                           |
> | ---------------------------------------- | ------------------------------ | ----------------------------------------------------------------------------------------------------------------------------------- | ---------------------------------------------------------- | ---------------------------------------------------------------------------------------------------- |
> | **SoP Judge Prompt**                     | Satisfaction of Products (SoP) | Determines whether a candidate product fully satisfies the user’s needs according to the rubric.                                    | (question, rubrics, model_product_name, model_response) | `<answer> Satisfied / Not Satisfied / Unable to Determine </answer>`                                 |
> | **Scenario Coverage Judge Prompt**       | Scenario Coverage              | Checks whether the agent-generated demands are semantically correct and collectively cover all requirements in the question.        | (question, rubrics, model_demands)                               | `<result> correct / missing / incorrect / fail </result>`                                            |
> | **Rationale Validity (RV) Judge Prompt** | Rationale Validity             | Determines whether the reasoning given by the model contains any factual inaccuracies when explaining why a product meets a demand. | (question, rubrics, model_demand, model_product_name, model_response)                             | `<result> 1 / 0 </result>` (1 = reasoning fully accurate; 0 = factual errors present)                |
> | **Safety-Critical Judge Prompt**         | Safety Rubric Pass Rate        | Checks whether the model’s response correctly identifies and addresses the safety trap described in the question.                   | (question, trap_rubric, model_response)                          | `<answer>correct</answer>` or `<answer>incorrect</answer>` (followed by a brief textual explanation) |

---

### Official Review · Reviewer_RBMc · 2025-11-03

**Soundness:** 2
**Presentation:** 3
**Contribution:** 2
**Rating:** 4
**Confidence:** 4

**Summary:**

This paper introduces ShoppingComp, a new benchmark for evaluating LLM-powered shopping agents. Its core contribution lies in the comprehensive evaluation of three key capabilities within a unified, open-web framework: 1) accurate product retrieval grounded in complex, real-world user requirements; 2) the quality of verifiable report generation, assessed against a detailed rubric; and 3) safety-critical decision-making in scenarios involving potential risks.

**Strengths:**

The paper presents a well-designed and highly challenging benchmark. Its preliminary findings provide valuable insights into the real-world limitations of current Large Language Models (LLMs), making a timely contribution to the field of agent evaluation.

**Weaknesses:**

1.The primary contribution leans more towards empirical findings rather than methodological innovation.
2.The Y-axis in Figure 4 lacks a clear label, which hinders the interpretation of the chart.
3.There is an inconsistency in the reported performance for GPT-5. The "Conclusion and Future Work" section states it "reaches only 19.6%", which appears to conflict with the data presented in Table 2. Please clarify or correct this discrepancy.

**Questions:**

1.Regarding the model's failures in safety-critical decision-making, could these be attributed to specific knowledge gaps in its pre-training data?
2. The paper commendably reports that the Report Verifier has only a 75.6% agreement rate with human annotators. This implies that nearly a quarter of the judgments in the report quality evaluation could be inaccurate. Could this level of uncertainty significantly affect the final ranking of the models and the validity of the conclusions drawn about their report generation capabilities?

---

> ### Author Response · Authors · 2025-11-26
> **To Reviewer RBMc**
>
> We sincerely thank the reviewer for the time and helpful feedback.
> > W1: The primary contribution leans more towards empirical findings rather than methodological innovation.
>
> ShoppingComp introduces a new evaluation paradigm for measuring an agent’s ability to both browse the web , produce decision supporting reports. It's a capability rarely explored in prior benchmarks such as WebShop or Mind2Web.  \
> Moreover, this evaluation paradigm is timely and urgently needed, as LLM-based shopping assistants are rapidly emerging in real-world use. OpenAI’s release *ChatGPT Shopping Research (https://openai.com/index/chatgpt-shopping-research)* on 24  November, highlights the same goal: product accuracy with many constraints. ShoppingComp provides the first open, academic framework to systematically evaluate these capabilities.
>
> >W2: The Y-axis in Figure 4 lacks a clear label, which hinders the interpretation of the chart.
>
> The X-axis label (“Average Time per Sample, minutes”) and Y-axis label (“Number of Samples”) have been added to Figure 4 to illustrate the time distribution between experts and annotators. This figure highlights the difficulty of ShoppingComp tasks and their associated human effort.
>
> >W3: There is an inconsistency in the reported performance for GPT-5. The "Conclusion and Future Work" section states it"reaches only 19.6%", which appears to conflict with the data presented in Table 2. Please clarify or correct this discrepancy.
>
> We have corrected a typo: GPT-5’s score is 11.22%, not 19.6%, consistent with Table 2. Many thanks.
>
> >Q1: Regarding the model's failures in safety-critical decision-making, could these be attributed to specific knowledge gaps in its pre-training data?
>
> We thank the reviewer for this insightful question. We conducted an ablation experiment under a no-tool setting, where models directly verified the Safety Critic Rubric using only pre-trained knowledge.
> As shown in the table, GPT-5 scored 57.7 % and Gemini-2.5-Pro 65.4 % when directly verifying the safety rubric under a no-tool condition. Yet, their actual Safety Rubric Pass Rates (65.4 % for GPT-5 vs. 46.2 % for Gemini-2.5-Pro) differ notably. This contrast shows that failures arise not only from lacking knowledge itself but also their ability to retrieve, reason, and response.
> | Model          | Safety Rubric Verification(Pre-trained Knowledge) | Actual Safety Rubric Pass Rate|
> | :----------------- | :-----------------------------: | :--------------------------------------------: |
> | GPT-5          |                             57.70 %                             |                     65.40 %                    |
> | Gemini-2.5-Pro |                             65.40 %                             |                     46.20 %                    |
>
> Further trajectory analysis supports this view.
> For example, “when using the microwave mode of a steam-bake combo oven, metal containers must not be used.” Gemini-2.5-Pro identified the potential safety risk during its reasoning process. GPT-5 obtained this warning from the *search_api* results and correctly included it in its final report. However, ChatGPT DeepResearch also retrieved the same warning from the *search_api* content but failed to mention it in its final report.
>
> These findings confirm that safety failures arise not only from pre-training data but also from search and report.
>
> >Q2: The paper commendably reports that the Report Verifier has only a 75.6% agreement rate with human annotators. This implies that nearly a quarter of the judgments in the report quality evaluation could be inaccurate. Could this level of uncertainty significantly affect the final ranking of the models and the validity of the conclusions drawn about their report generation capabilities?
>
> This is an important concern and appreciate the reviewer highlighting it. Firstly, each individual rubric exceeds 85 % human-LLM agreement and the combined rate with multiple rubrics is 75.6 %. The combined rate higher than comparable benchmarks such as HealthBench (70.9%) and Research Rubrics (76%), confirming robustness.
> To further ensure reliability:
>   - For product retrieval, we also rely on ground-truth AnswerMatch-F1, a fully human-verified metric.
>   - For report evaluation, we average multiple rubric-based judgments to mitigate single verifier noise in Appendix D.
>  Across three judges (Gemini-2.5-Pro, GPT-4o, DeepSeek-V3.1), model rankings remain consistent, supporting the validity of our comparative conclusions.
>
> **Summary**
>
> We sincerely thank the reviewer once again for the thoughtful and constructive feedback. We have carefully addressed the concerns through clearer dataset descriptions, additional analyses, and expanded discussions on safety reasoning and evaluation reliability. We hope these updates can further strengthen the paper.

---

### Official Review · Reviewer_QH8m · 2025-11-03

**Soundness:** 3
**Presentation:** 2
**Contribution:** 3
**Rating:** 4
**Confidence:** 4

**Summary:**

This paper introduces ShoppingComp, a practical benchmark for evaluating LLM-based shopping agents along three axes: (i) Browse Products, (ii) Expert-level Report Generation, and (iii) Safety-Critical Decision Making. The benchmark contains 120 tasks and 1,026 scenarios.
The evaluation of this benchmark uses LLM-as-a-Judge (e.g., Gemini-2.5-Pro), and the results show that LLM-based agents have large gaps to humans (e.g., GPT-5 AnswerMatch-F1 11.22% vs expert 25.73%).
This paper focuses on a practical application scenario of LLM-based agents, but it misses many details of data collection and categorization. This makes the paper less suitable for the dataset/benchmark area and more like a study that reveals problems and gaps.

**Strengths:**

1. Interesting and practical topic. This paper focuses on a valuable and interesting application scenario, namely, LLM-based shopping agents.
2. Empirical takeaways from the experiments. The paper conducts large-scale experiments to highlight the bottlenecks of LLM-based agents in real-world deployment.
3. Well-defined scoring dimensions. The scoring includes three parts of evaluation metrics, which provide a comprehensive foundation for future research.

**Weaknesses:**

1. Dataset accessibility and details. The paper promises future open-sourcing but provides no current release, subset, or several appendix samples; this undermines claims of reproducibility for a benchmark/dataset paper.
2. Potential evaluation bias risk. Gemini-2.5-Pro is used as both the LLM evaluator and the evaluation target, raising potential bias concerns (existing research has revealed that LLM evaluators would prefer the results generated by themselves)
3. Poor organization. As a dataset/benchmark paper, this paper spends less than 1.5 pages on the dataset construction and omits many details directly related to the fairness and reliability of the dataset.
4. Missing References, e.g., [1].
5. Potential copyright issue. The authors directly use real-world product images and even links. One of the concerns is the author's data sources, the licenses obtained, and whether there are any copyright risks.

**Questions:**

1. Missing Dataset Details:
   1. The authors claim they spent at least 4,000 person-hours building the dataset, but the dataset is neither open-source nor included in supplementary materials, and even a portion of their dataset is not shown in the appendix. Worse still, one of the highlights, safety questions/safety traps (e.g., in Table 1), lacks details. What types of safety traps are included? Are they designed based on existing work or just from ad-hoc ideas? Do safety traps include common dark patterns in shopping (e.g., concealing key parameters in product descriptions, using keywords easily confused with other popular products, etc.)?
   2. Where are the `real e-commerce queries` analyzed in the user questions from (Line 230)? How are `diverse, purpose-driven user prompts` synthesized? Could this process introduce bias?
   3. Line 239: What is the criterion of an `overly easy question`?
   4. Line 269 mentions `high-value categories such as home appliances, electronics...`. After searching with these keywords, I realize that the authors have not mentioned how many categories this paper includes, nor the meaning and scope of the categories, like `home appliances`. Furthermore, which categories are high-value? Which are low-value?
   5. Too many details about the dataset remain unknown. These details are crucial for understanding potential copyright, bias, and other issues related to the benchmark. Even if the main text cannot accommodate them, this content should at least be included in supplementary materials or an appendix.
2. In real-world shopping, there may be multiple products that meet the users' requirements. Does the dataset include this situation? How to determine the ground truth in this situation?
3. Evaluation Risks: Section 4.1.2 mentions that this paper uses LLM-as-a-judge to evaluate the quality of product recommendation reports. Is the Gemini-2.5-Pro ​model ​used here? Does this model also appear as an evaluation object in Table 2? Existing research has shown that LLM-as-a-judge is more likely to give its own generated content a high score [1]. One concern is that such an evaluation setting may introduce bias and reduce the reliability of the observation results.
4. Missing Reference: What are the differences compared to the Product Comparison Dataset for Online Shopping dataset [2] in the same field? Does this paper use the data samples from this dataset?
5. Potential Copyright Risks: The authors directly used real-world product images and even links (as shown in Fig. 2). One concern is the authors' data sources. Did the authors obtain permission from the data sources (and what kind of permission), and are copyright risks involved in this paper?


[1] Llm evaluators recognize and favor their own generations. Advances in Neural Information Processing Systems. 2024.

[2] Generating explainable product comparisons for online shopping. Proceedings of the Sixteenth ACM International Conference on Web Search and Data Mining. 2023.

---

> ### Author Response · Authors · 2025-11-26
> **To Reviewer QH8m (1/2)**
>
> We sincerely thank the reviewer for the time and helpful feedback.
>
> **1. Dataset details (W1 and Q1)**
> > W1: Dataset accessibility and details. The paper promises future open-sourcing but provides no current release, subset, or several appendix samples; this undermines claims of reproducibility for a benchmark/dataset paper. \
> Q1: Missing Dataset Details.
>
> We have added a complete *monitor* example in Appendix B , showing user question, rubrics, ground-truth products, and evidence traces. We also updated Figure 3(workflow) to present the data construction process more clearly.
>
> - **Dataset accessibility** Our dataset will be publicly released in huggface, in December 2025.
> - **Safety trap details.** The paper defines two main categories:
>
>   1. *Physical or health risks* — e.g., using metal trays in microwave ovens, or skincare products with potential allergic triggers.
>   2.  *Information level risks* — e.g., misleading efficacy claims. Our safety traps only cover problems identifiable through online information. Cases that require physical experience or post-purchase inspection to detect false advertising are not covered by ShoppingComp.
>
> - **User question synthesis.** No real user data are used. We have renamed *“User prompt”* to *“Synthesized user prompt”* (L230). The pipeline  Figure 3 (pp. 4) starts from 100,000 seed products with high transaction value and complex decision attributes. An agent system generates diverse, purpose-driven user questions, followed by three rounds of expert or annotator review. To ensure realism, we compared these questions with 1,000 real queries collected by human, from public shopping forums (e.g., Amazon Review ) and confirmed distributional alignment.
>
> - **Definition of “overly easy question.”**
>   1. Remove questions with too many valid products (#product > 10).
>   2. Remove easy questions correctly answered by 60% or more of the agents.
>
> - **Category details.** Section 3.4 now lists ten categories following Amazon’s taxonomy. “High-value” categories refer to those with higher average transaction prices or multi-attribute/safety-critical decisions (e.g., *appliances, electronics*).
>
> - **Copyright clarification.** All data source and licensing (CC0) details  was added.
>
>
> &nbsp;
>
> **2. Potential evaluation bias risk (W2 and Q3)**
> > W2: Potential evaluation bias risk. Gemini-2.5-Pro is used as both the LLM evaluator and the evaluation target, raising potential bias concerns (existing research has revealed that LLM evaluators would prefer the results generated by themselves) \
> Q3: Evaluation Risks.
>
> We appreciate this important observation. We acknowledge that LLM-as-a-judge evaluation may introduce self-preference bias, as discussed in ResearchRubrics (Scale AI 2025). To mitigate this:
> - *Human-grounded evaluation.* Precisely because of this concern, one of ShoppingComp’s key advantages is that, we does not rely solely on LLM judgment. We invested substantial human effort in constructing ground-truth product lists through multi-path retrieval and manual verification. Model performance is primarily assessed using the AnswerMatch-F1 metric against these human-verified ground truths., providing a fair and reproducible foundation independent of any single LLM judge’s preference.
> - *Multi-judge ablation.* We evaluated with three judges : Gemini-2.5-Pro, GPT-4o, and DeepSeek-V3.1 in appendix D. While absolute scores vary, model rankings remain consistent. Notably, under the Gemini-2.5-Pro judge, GPT-5 (50.13 %) outperforms Gemini-2.5-Pro (47.79 %), and this pattern holds across other judges (GPT-4o: 46.23 vs 43.44 %; DeepSeek: 38.55 vs 36.13 %), indicating minimal self-preference.
> | **Model**                    | **Gemini-2.5-Pro Judge** |   **GPT-4o Judge**   | **DeepSeek-V3.1 Judge** |
> | :---------- | :---------: | :------: | :------: |
> | **Gemini-2.5-Pro**           |     47.79 % ± 3.94 %     |   43.44 % ± 3.33 %   |     36.13 % ± 3.43 %    |
> | **DeepSeek-V3.1 (Thinking)** |     37.00 % ± 1.97 %     |   35.33 % ± 2.72 %   |     30.59 % ± 2.74 %    |
> | **GPT-5**                    |   **50.13 % ± 3.42 %**   | **46.23 % ± 4.06 %** |   **38.55 % ± 2.30 %**  |
>
>
> &nbsp;
>
> **3. Organization and Dataset Construction Clarity (W3)**
> >W3: Poor organization. As a dataset/benchmark paper, this paper spends less than 1.5 pages on the dataset construction and omits many details directly related to the fairness and reliability of the dataset.
>
> We have redrawn the workflow (Figure 3) with four explicit stages and corresponding examples, and added complete prompt templates in Appendix C to illustrate LLM-as-a-Judge evaluation. These improvements make the pipeline transparent and reproducible.

---

> ### Author Response · Authors · 2025-11-26
> **To Reviewer QH8m (2/2)**
>
> **4. Missing references (W4 and Q4)**
> >W4: Missing Reference: What are the differences compared to the Product Comparison Dataset for Online Shopping dataset [2] in the same field? Does this paper use the data samples from this dataset?\
> >Q4: Missing Reference
>
> We are grateful for this correction and have added the missing reference (Generating Explainable Product Comparisons for Online Shopping, WSDM 2023). We apologize for not citing it earlier. We briefly summarize the distinction between that work and ours in the revised version.
>
> &nbsp;
>
> **5. Potential copyright issue (W5 and Q5)**
> >W5: Potential copyright issue. The authors directly use real-world product images and even links. One of the concerns is the author's data sources, the licenses obtained, and whether there are any copyright risks.\
> >Q5: Potential Copyright Risks
>
> We sincerely thank the reviewer for this reminder. As mentioned in Q1, We has confirmed that all data usage complies with relevant copyright and data-protection laws. The open-sourced dataset will be released under a CC0 license, containing only original product URLs and direct image links, without any redistribution or modification. To ensure long-term usability, we manually summarize key factual information (e.g., attributes, specifications) from each source, preserving reproducibility while maintaining full legal compliance.
>
> &nbsp;
>
>
> **6. Many products meet (Q2)**
> >Q2: In real-world shopping, there may be multiple products that meet the users' requirements. Does the dataset include this situation? How to determine the ground truth in this situation?
>
> This is an excellent question and indeed a central consideration in the design of ShoppingComp, which required significant human verification effort. This challenge motivated the design of AnswerMatch-F1 and Recall metrics. In real-world markets, some products have many substitutes, while others are unique due to their functions or features. Thus, while the ground-truth list may contain multiple products, experts are instructed to design tasks with fewer than ten valid items to ensure tractable evaluation.
>  During user-question synthesis, we prioritize seed products that feature distinctive technologies or attributes to minimize overlap. Our multi-source retrieval and filtering process guarantees that the final product set for each question remains concise and verifiable.
>
> The recall metric was introduced precisely for this reason: in real shopping scenarios, users not only need to find products that meet their needs but also to compare these products to make final purchase decisions. Effective agents must therefore first retrieve all valid options. As shown in Table2(pp. 9), current models still struggle with recall, which in turn limits their ability to perform reliable product comparisons and informed decision-making.
>
> &nbsp;
>
> **Summary**
>
> We thank the reviewer again for these valuable comments. The revised version includes:
> - Expanded dataset and category details.
> - Clarified safety-trap taxonomy.
> - Multi-judge ablation addressing evaluation bias.
> - Added missing references.
> - Verified copyright compliance(CC0).
> - Recall metric design for many-products problem.
>
> We sincerely hope these substantial revisions address your concerns and will lead you to reconsider your evaluation.

---

### Official Review · Reviewer_u67A · 2025-11-04

**Soundness:** 1
**Presentation:** 2
**Contribution:** 1
**Rating:** 2
**Confidence:** 4

**Summary:**

The authors present a large data set they compiled to evaluate LLMs for several online shopping tasks, such as product retrieval, safety-related recommendation, scenario coverage, etc. They describe some of the tasks and how the data was collected and curated, and provide initial evaluation of several strong LLM baselines.

**Strengths:**

- Very relevant and timely problem being considered.
- Large data set curated and to be publicly release.
- Interesting new tasks considered, especially around safety.
- Promising results shown, and some interesting patterns uncovered.

**Weaknesses:**

- Very poorly written at times, too high level.
- Handwavy explanations, reads like a quick summary without actual deep explanations.
- Evaluation is also high-level and handwavy, without examples to help understand the data the tasks in a deeper manner.

**Questions:**

- Figures 1 and 5 are never referenced in the text.
- Many of the explanations are very high-level, without clarifying examples and case studies. E.g., line 88, "typically express needs in ...", some examples would help immensely. This lack of clear and reasonable examples repeats throughout the work.
- Are there any X's in Table 1? When such tables with checkmarks are provided, it is always important to see what are the downsides.
- The authors provide some examples of the data, but at an insufficient level. More discussion around the data and how it looks like should be added.
- In addition, data statistics are insufficiently discussed.
- Line 124, what do these number mean, on what data, on what task? Very unclear.
- Similarly, examples or deeper discussion missing in lines 234, and in 239.
- Also in line 250.
- The results in this paragraph are just given, without deeper discussion, making it very difficult to appreciate the results.
- Line 254, "the verifier evaluates report quality against ...", how exactly? Very vague.
- Line 442, "three patterns emerge", some examples?

Overall, the work is written at just a too high a level, without an attempt to actually help the readers understand what is the data actually being used. It reads as a brief and a high-level summary, that might be sufficient for a webpage for example, but far from sufficient for a venue such as ICLR.

---

> ### Author Response · Authors · 2025-11-26
> **To Reviewer u67An (1/3)**
>
> **Summary**
>
> We sincerely thank the reviewer for the time and this constructive feedback. In response to high-level and handwavy explanations, we have replaced all previously high-level descriptions with explicit data examples and quantitative analyses in the revision. The manuscript has been updated, and all modifications are highlighted in color (blue for revisions, red for additions).
>
> 1. **More detailed step-by-step data collection process.** Section 3.2 and Figure 3 （pp. 4）now provides a detailed, four-stage human-in-the-loop pipeline covering question construction, rubric generation, multi-source product retrieval, and expert-verified instance filtering. Each step includes concrete examples (e.g., find a new mouse for the user) and explicit filtering criteria (e.g., >10 valid products, easy questions, semantic duplicates) to replace previous abstract descriptions. Section 3.4 adds dataset statistics (category distribution, expert vs. annotator time) and defines “high-value” categories to illustrate why tasks are intrinsically difficult.
> 2. **Concrete examples.** We adds two complete data cases (monitor and microwave safety-trap tasks) in  Appendix B (pp. 14–15), showing questions, rubrics, ground-truth products with evidence, and actual tool-call traces from Gemini-2.5-Pro and GPT-5, allowing readers to see how the benchmark operates end-to-end and how errors arise in practice.
> 3. **LLM-as-a-Judge prompt templates.** We now include the full prompt templates for LLM verifiers, clarifying how evaluation is performed at the rubric and evidence level in Appendix C (pp. 18–20).
> 4. **Clearer evaluation discussion and ablations.** Section 4.4 has been added to thoroughly analyze the tool's usage and resource efficiency. We provide a concise discussion of Figure 1 in Section 1 to explain the overall performance of different agents and LLMs on ShoppingComp.
>
> We have also carefully polished the overall writing for clarity and examples, and we sincerely hope these substantial improvements will lead you to re-evaluate our work in light of the new evidence.
>
> &nbsp;
>
> >Q1: Figures 1 and 5 are never referenced in the text.
>
> We thank the reviewer for highlighting this missing clarification of Figures 1 and 5.
>
> *1. Key findings from Figure 1:*
> The four metrics in Figure 1 map directly to the three core capabilities that ShoppingComp evaluates:
> - **AnswerMatch-F1** → precise product retrieval (Top-left) ：All models still perform poorly on precise retrieval (<20% F1). This remains the hardest part of the pipeline, requiring real-world product knowledge + web retrieval + multi-constraint reasoning. GPT-5 is 2–3× stronger than others and substantially better than GPT-4o, suggesting clear benefits from tool-use and deliberate reasoning.
> - **Scenario Coverage-F1** →completeness report generation  (Top-right) Coverage scores are generally high, but models such as GPT-4o still miss essential constraints (e.g., wattage, size, compatibility), explaining why reports can look correct but lead to wrong product recommendations.
> - **Report Rationale Validity** → hallucination-free report generation (Bottom-left) Rationale Validity is relatively strong across models, showing progress in hallucination reduction, though Claude remains less grounded.
> - **Safety Rubric Pass Rate** → safety critical decision making (Bottom-right) Safety lags furthest behind real world standards. GPT-5 (65.4%) is still far from deployment ready, showing that safety-critical reasoning remains an open problem.
>
>  *2. Key findings from Figure 5:*
>
>  Figure 5 follows Amazon’s official taxonomy, showing ten major categories: Appliances (37%), Home & Kitchen (18%), Electronics (16%), Beauty & Personal Care (14%), and others. This alignment ensures domain balance and realistic diversity, emphasizing high-price, high-involvement categories, where consumers face complex trade-offs and longer decision cycles.

---

> ### Author Response · Authors · 2025-11-26
> **To Reviewer u67An (2/3)**
>
> >Q2: Many of the explanations are very high-level, without clarifying examples and case studies. E.g., line 88, "typically express needs in ...", some examples would help immensely. This lack of clear and reasonable examples repeats throughout the work.
>
> We agree that the paper would benefit from more concrete examples. In the revised version, we add two illustrative case studies in the Appendix B (pp. 14–15, *“a microwave oven that cannot heat metal containers”* and *“selecting a monitor for a game live streamer”*) , to make the dataset easier to understand.
> Regarding line 88 (“users typically express needs in…”), for example, a typical user query is:
>
> *“I need a phone that takes great photos.”*
>
> rather than the specification-level form:
>
> *“Recommend a smartphone with a 1/1.3'' sensor, >F1.7 aperture, and OIS support.”*
>
> This “natural-language intention vs. specification-level requirement” gap is precisely what makes accurate retrieval and reasoning difficult.
>
>
> >Q3: Are there any X's in Table 1? When such tables with checkmarks are provided, it is always important to see what are the downsides.
>
> Unlike OPeRA, it does not include user personas. This X has been added to Table 1 in the revised version.
>
>
> >Q4: The authors provide some examples of the data, but at an insufficient level. More discussion around the data and what it looks like should be added.\
> >Q5: In addition, data statistics are insufficiently discussed.
> - **Examples of data.** We have added a detailed data example (the monitor case) in Appendix B.2 (page 14), including the dataset inputs (question,rubrics, ground-truth) and tool call traces of two models: Gemini-2.5-Pro and GPT-5.
> - **Data statistics.** In addition, we have substantially enriched the dataset analysis to address this concern.  As mentioned in Q1, the revised paper now clearly specifies that ShoppingComp contains 10 product categories following the Amazon taxonomy, and explicitly defines high-value categories. High-value categories, for example, a washing machine requires balancing performance, energy efficiency, and installation constraints across different usage scenarios.
> - **Human Performance.** We also provide quantitative time statistics: experts spend on average 24.6 minutes per case and annotators 52.3 minutes, after capping durations longer than one hour (experts) or two hours (annotators). In contrast, the best-performing model (GPT-5) completes the same tasks in about 3 minutes on average. These results show that ShoppingComp is hard enough to  measure a model’s retrieval and resoning capability, while  valuable for high human time costing shopping tasks.
>
> >Q6: Line 124, what do these number mean, on what data, on what task? Very unclear.
>
> The numbers in Line 124 in introduction refer to the AnswerMatch-F1 metric (refer to 4.1.1) of two models(GPT
> -5, Gemini-2.5-Flash) and human performance. The AnswerMatch-F1 measures precise product retrieval, the core evaluation task in our benchmark. We have restored the full clarification in the revised version.
>
> >Q7: Similarly, examples or deeper discussion missing in lines 234, and in 239.
>
> To address the missing details in lines 234 and 239 on ground-truth products and evidence generation and instance filtering steps, we have expanded and clarified these steps as follows (revision the Figure 3 pp.4):
> - *Step 3: Multi-source Candidate Product Construction.*
>
> Candidate products are retrieved via a hybrid multi-source process combining web-agent search, expert-curated additions, and embedding-based similarity expansion. Annotators then apply multi-stage manual filtering with evidence verification—cross-checking each item’s official specs, user reviews, and images.
> In the gaming mouse example, 34 initial candidate mice were gathered through multi-source retrieval, expert cross review, and validation.
>
> - *Step 4: Ground-truth Verification and Instance Filtering*
>
> Annotators verify each candidate against the rubric and attach supporting evidence：official specs, images, and review excerpts, to build the ground-truth set. In the gaming mouse example, 34 initial candidate  were finally reduced to 3 fully verified mice as ground-truth products after multi-stage human filtering.
>  A layered filtering process then removes
> - questions with > 10 valid products
> - easy ones solved by 60% of  models
> - semantically duplicated queries detected via embedding clustering.
> Finally, experts cross-review remaining instances to ensure accuracy, fairness, and diversity, yielding a realistic and rigorously validated benchmark.

---

> ### Author Response · Authors · 2025-11-26
> **To Reviewer u67An (3/3)**
>
> >Q8: Also in line 250.\
> >Q9: The results in this paragraph are just given, without deeper discussion, making it very difficult to appreciate the results.
>
> It indeed concerns a key agreement-rate metric for evaluating the reliability of our LLM-as-a-Judge framework. In our benchmark, the agreement rate reached 81% at the scenario level (across 1,026 scenarios) and 84% at the question level (across 120 tasks). These results demonstrate the high alignment between human and automated evaluations in ShoppingBench, exceeding those reported by HealthBench (70.9%)[1] and ScaleAI’s ResearchRubric (76%)[2]. Minor inconsistencies primarily stem from human annotation noise and occasional retrieval or reasoning errors, rather than systematic disagreement.
>
> >Q10: Line 254, "the verifier evaluates report quality against ...", how exactly? Very vague.
>
> To make the evaluation transparent and reproducible, ShoppingComp implements an LLM-as-a-Judge verification framework composed of four complementary prompts, each targeting a distinct quality dimension of model-generated reports.
> | **Prompts**                             | **Metric**                     | **Purpose**                                                                                                                         | **Input Format**                                           | **Output**                                                                                           |
> | ----------- | -------- | -------------- | -------------- | ------------ |
> | **SoP Judge Prompt**                     | Satisfaction of Products (SoP) | Determines whether a candidate product fully satisfies the user’s needs according to the rubric.                                    | (question, rubrics, model_product_name, model_response) | `<answer> Satisfied / Not Satisfied / Unable to Determine </answer>`                                 |
> | **Scenario Coverage Judge Prompt**       | Scenario Coverage              | Checks whether the agent-generated demands are semantically correct and collectively cover all requirements in the question.        | (question, rubrics, model_demands)                               | `<result> correct / missing / incorrect / fail </result>`                                            |
> | **Rationale Validity (RV) Judge Prompt** | Rationale Validity             | Determines whether the reasoning given by the model contains any factual inaccuracies when explaining why a product meets a demand. | (question, rubrics, model_demand, model_product_name, model_response)                             | `<result> 1 / 0 </result>` (1 = reasoning fully accurate; 0 = factual errors present)                |
> | **Safety-Critical Judge Prompt**         | Safety Rubric Pass Rate        | Checks whether the model’s response correctly identifies and addresses the safety trap described in the question.                   | (question, trap_rubric, model_response)                          | `<answer>correct</answer>` or `<answer>incorrect</answer>` (followed by a brief textual explanation) |
>
> >Q11: Line 442, "three patterns emerge", some examples?
>
>  We give three examples for  the three patterns:
> - **Persuasive but narrow recall.** As shown in Appendix B.2, in the Gemini generated report for the monitor case, the model stopped retrieval after finding a single monitor (KTC 32-inch 4K Mini LED Monitor), failing to continue searching for other valid candidates. This led to a persuasive yet incomplete report with low recall.
> - **Lack of comparison.** To find a wireless mouse for the user in Figure 3, GPT-5 retrieved multiple similar products but failed to compare them effectively. For instance, between Razer Viper V3 Pro and Razer Viper V3, GPT-5 incorrectly assumed both were wireless mice, overlooking that V3 Pro is wireless while V3 is wired. An error caused by insufficient comparison.
> - **Inconsistent safety reminders.** In the ChatGPT Deep Research case, the model produced a fluent and detailed recommendation report but omitted a key warning that metal baking trays must not be used in microwave mode.
>
> [1] Arora, S. et al. (2024). HealthBench: Evaluating LLMs in Clinical Reasoning. OpenAI Technical Report
>
> [2] Scale AI (2025). Research Rubrics: A Framework for LLM-as-a-Judge Reliability Evaluation. arXiv:2511.07685

---

### Author Response · Authors · 2025-11-27
**Global Response**

We sincerely thank all reviewers for their time and for the constructive, encouraging feedback. We deeply appreciate the recognition of ShoppingComp as a timely, interesting, well-designed, and highly challenging benchmark for evaluating LLM-based shopping agents. We are also grateful that reviewers emphasized the work’s key strengths: its real-world relevance, comprehensive evaluation design, and focus on safety-critical decision making.

In this revision (blue for revisions, red for additions), we have carefully addressed all concerns across reviews :

- Clarity and depth. We replaced high-level summaries with step-by-step procedural details, quantitative analyses, and full case studies (Appendix B–C), making the dataset construction and evaluation pipeline transparent and reproducible.
- Dataset accessibility and copyright. The dataset will be released on Hugging Face under a CC0 license, accompanied by the full evaluation toolkit on GitHub for community use.
- Evaluation fairness. We conducted multi-judge ablations using Gemini-2.5-Pro, GPT-4o, and DeepSeek-V3.1, and found that model rankings remain consistent across all judges. Model performance is primarily assessed using the AnswerMatch-F1 metric computed against human-verified ground truths, which we regard as one of the core strengths of ShoppingComp.
- Analytical depth. New ablation studies on tool usage, resource efficiency, and human performance (Sec. 4.4) provide deeper insights into retrieval, reasoning, and reporting behavior.

Overall, ShoppingComp offers several unique advantages:

- **Real-world, open-web retrieval**, evaluating agents on authentic, verifiable products rather than closed catalogs.
- **Multi-constraint problem formulation**, reflecting realistic consumer decision-making under complex trade-offs.
- **High-quality, multi-round human and expert annotation**, ensuring reliability, diversity, and reproducibility.
- **Comprehensive safety evaluation**, systematically assessing whether models can avoid unsafe or misleading product recommendations.

Interestingly, OpenAI’s recent *ChatGPT Shopping Research* (Nov 2025, [link](https://openai.com/index/chatgpt-shopping-research/)) adopted a similar evaluation setup: *“a new evaluation composed of difficult product discovery queries with many constraints,”* measuring *product accuracy* as the percentage of recommended items that meet user requirements (e.g., price, color, material, specs).
 This metric conceptually corresponds to the *Satisfaction of Products (SoP)* metric in ShoppingComp. While OpenAI’s study demonstrates the industrial relevance of this evaluation paradigm, ShoppingComp formalizes it into an open, standardized, and rigorously validated benchmark, with multi-constraint questions, verified evidence, and reproducible evaluation. It bridges the gap between practical deployment needs and academic benchmarking rigor. We are actively testing our benchmark under this new *ChatGPT Shopping Research* framework and will include updated comparative results in the camera-ready version.

We believe ShoppingComp fills a critical evaluation gap for LLM-based shopping agents, offering a reproducible foundation for future research on retrieval-augmented reasoning, safety, and real-world deployment. We sincerely hope the reviewers will reconsider our work in light of the substantial improvements and new supporting evidence presented in this revision.

---

### Note · Program_Chairs · 2026-01-17
**Submission Desk Rejected by Program Chairs**

The following references in this submission do not refer to real documents and/or have major errors in bibliographic information:

 Haoran Luo, Yusen Zhang, Haoran Yu, et al. Api-bench: Evaluating llms on function calls. arXiv preprint arXiv:2311.09816, 2023.